# Lymphoepithelioma-like Carcinoma of the Breast Synchronous with a High-Grade Invasive Ductal Carcinoma and Ductal Carcinoma in Situ in a Different Quadrant of the Same Breast: A Case Report

**DOI:** 10.3390/medicina58091146

**Published:** 2022-08-23

**Authors:** Vasil Nanev, Silvia Naneva, Angel Yordanov, Strahil Strashilov, Assia Konsoulova, Mariela Vasileva-Slaveva, Tatyana Betova, Ivan Ivanov

**Affiliations:** 1Department of Surgical Oncology, Dr. Georgi Stranski University Hospital, Medical University—Pleven, 5800 Pleven, Bulgaria; 2Hematology Clinic, Dr. Georgi Stranski University Hospital, Medical University—Pleven, 5800 Pleven, Bulgaria; 3Department of Gynaecologic Oncology, Medical University—Pleven, 5800 Pleven, Bulgaria; 4Department of Plastic, Reconstructive and Aesthetic Surgery, Dr. Georgi Stranski University Hospital, Medical University—Pleven, 5800 Pleven, Bulgaria; 5USHATO—Sofia National University Cancer Hospital, 1000 Sofia, Bulgaria; 6Women for Oncology, 1000 Sofia, Bulgaria; 7Department of Preclinical and Clinical Disciplines, Faculty of Social Health and Healthcare, University Prof. A. Zlatarov, 8000 Burgas, Bulgaria; 8Department of Breast surgery, Shterev Hospital, 1000 Sofia, Bulgaria; 9Research Institute, Medical University—Pleven, 5800 Pleven, Bulgaria; 10Department of Clinical Pathology, Dr. Georgi Stranski University Hospital, Medical University—Pleven, 5800 Pleven, Bulgaria

**Keywords:** lymphoepithelioma-like breast carcinoma, tumor heterogeneity, NST, breast cancer, DCIS

## Abstract

Lymphoepithelioma-like breast carcinoma (LELC) is a rare type of malignant breast tumor that is not included in the current edition of the World Health Organization (WHO) classification of breast tumors. Currently, there are no clearly defined therapeutic strategies, and the general information on breast LELC is based on sporadic clinical cases described in the medical literature. We present a clinical case that describes a 49-year-old woman with a tumor formation in the right breast, histologically verified as LELC, together with a non-palpable, synchronous high-grade invasive ductal carcinoma and ductal carcinoma in situ Grade 2 (DCIS G2) in a different quadrant of the same breast. To our knowledge, this is the first case described in the literature that combines a LELC with a synchronous carcinoma in the same breast.

## 1. Introduction

Lymphoepithelioma-like breast carcinoma (LELC) is a rare type of malignant tumor that is not included in the current edition of the World Health Organization (WHO) classification of tumors of the breast [1]. It can be found in one of the previous editions of the series [2].

Presently, there are no clearly defined therapeutic strategies, and the general information concerning breast LELC is based on the clinical cases described in the medical literature. Kumar and Kumar described the first case of breast LELC in 1994 in a 65-year-old woman [3]. The tumor was histologically identical with the nasopharyngeal lymphoepithelioma described by Regaud Reverchon [4] and Schminke [5] in 1921. Over the years, other anatomical locations of LELC have been reported, such as the oesophagus [6], larynx [7], lungs [8], stomach [9], thymus [10], trachea [11], bladder [12], ovary [13], cervix [14], skin [15] and thyroid [16]. Some anatomical sites and cases of LELC have been reported to be associated with the Epstein–Barr virus (EBV) [17] and human papilloma virus (HPV) [18].

Despite the fact that since 2012, the WHO classification of the breast tumors has not included LELC [19], in the literature review, we found 39 cases of LELC of the breast, reported up until 2021 [3,20,21,22,23,24,25,26,27,28,29,30,31,32,33,34,35,36,37,38,39,40,41,42], We present a clinical case of a 49-year-old woman with a tumor formation in the right breast, histologically verified as LELC. We add another case to the existing cases in the literature, which is different in terms of the additional finding in the form of a synchronous tumor in the same breast.

## 2. A Case Report

We report a case of a 49-year-old woman with a right breast tumor lesion, histologically proven as LELC, together with a synchronous non-palpable high-grade invasive NST (no special type) ductal carcinoma and ductal carcinoma in situ Grade 2 (DCIS G2) [1] in a different quadrant of the same breast. The patient signed informed consent before performing the diagnostic and treatment procedures.

In February 2020, the patient was admitted to the surgical clinic for diagnostic work-up. She reported having found a palpable mass in the right breast one month ago, which increased in size rapidly. With regard to previous surgical history, she reported having undergone surgery for uterine leiomyomas and ovarian cysts (with total hysterectomy and adnexectomy) five months prior to the current period of hospitalization. No other co-morbidities were reported.

A mammography, performed at the end of January 2020, showed a strong lobulated shadow in the lower outer quadrant of the right breast with a size of 43/27 mm. The lesion was classified as BIRADS 4 (Figure 1).

The breast ultrasound showed an anechogenic formation with smooth and sharp outlines of irregular shape; there was a distal signal of amplification similar to cystic formation. The mammography and breast ultrasound did not show any other lesions or abnormalities.

Fine-needle biopsy (FNB) under ultrasound control was performed, and the material was sent for cytological examination. The result showed a compact protein structure, erythrocytes, scarce fibrosis and single ductal cells in the normal tissue, and small strands of leukocytes.

The physical examination revealed a palpable formation in the lower lateral quadrant of the right breast, with a size of about 40–45 mm, with a dense-elastic consistency, which was relatively mobile and painless. The overlying skin was intact. No other solid lesions were palpated in either mammary gland; there were no pathologically enlarged lymph nodes in either axilla. Upon hospital admission, a chest X-ray showed no pathology. An excisional breast biopsy was performed under general anesthesia. The resected formation was cut before fixation, presenting a non-encapsulated, circumscribed, nodular tumor 45/30 mm in size, with a dense-elastic consistency (Figure 2).

The fixed preparation was sent for histological examination, together with the complete documentation.

The histological results demonstrated breast parenchyma with tumor infiltration, represented by cells with moderate to pronounced pleomorphic vesicular nuclei with coarse-grained chromatin, obvious nucleoli and non-abundant bright cytoplasm (Figure 3).

Tumor cells grow in groups and strands in a pronounced lymphocyte stromal response. T-lymphocytes occupy approximately 30% of the stroma area. From the immunohistochemical examination (IHC), ER, PgR and HER2 were negative, Ki-67 was positive in about 50% of the tumor cells, E-cadherin demonstrated moderately intense staining of cell membranes in 70% of the tumor cells and p63 nuclear positivity in about 20% of the tumor cells. CK 7 was strongly positive in about 40% of the tumor cells, CK 5/6 staining was weak in 2% of the tumor cells, vimentin was strongly positive in the tumor cells, CD 117 demonstrated weak staining of the cytoplasm in 5% of the tumor cells and CD20 and CD3 were positive in B lymphocytes and T lymphocytes, respectively (Figure 4). Pd-L1 staining was not carried out.

The morphological and IHC data report a high-grade poorly differentiated carcinoma with pronounced tumor-infiltrating lymphocytes and these findings are in accordance with LELC.

For complete staging of the disease, a CT of the whole body was carried out and no distant metastasis was reported. The final stage was pT2 cN0 cM0 G3. The case was discussed at a session of the multidisciplinary tumor board and the patient was referred to surgery; radical mastectomy with axillary lymph node dissection was performed one month after the initial diagnosis. The result of the histopathological examination in the tumor bed showed no residual tumor. An in situ lesion, recognized as DCIS G2 and a focus of invasive NOS ductal carcinoma G3, with pronounced tumor-infiltrating lymphocytes, was diagnosed in another quadrant of the breast (Figure 5).

The examined 17 axillary lymph nodes had pronounced sinus histiocytosis. IHC of the invasive ductal carcinoma (IDC) demonstrated weak positive staining for both steroid receptors; ER and PgR each demonstrated weak nuclear staining in about 10% of the tumor cells. HER2 was assessed as 2+ moderate, complete membrane staining. The Ki-67 index was 30% and E-cadherin demonstrated moderately intense staining of the cell membranes of the tumor cells (Figure 6). Subsequently, HER2 demonstrated a negative status regarding in-situ hybridization.

The patient refused chemotherapy, and the adjuvant therapy continued with anti-estrogen treatment. Twenty-five months after the surgery, the patient has no evidence of local or distant recurrence of the disease.

## 3. Discussion

The described case of lymphoepithelioma-like carcinoma of the breast, coexisting with NOS ductal carcinoma, demonstrated a morphological and immuno-profile of the lymphoepithelioma-like component, which is consistent with most of the cases of lymphoepithelioma-like carcinomas previously described in the literature [26,37,40,41,42]. The presence of two heterogeneous components demonstrated variation in ER, PgR, HER2 and Ki-67 expression, making prediction of the treatment response and survival prognosis difficult. According to the currently available guidelines, different treatment approaches are required for the two coexistent lesions in the breast [43]. The mentioned circumstances make such cases difficult with regard to diagnostic evaluation and treatment [44,45]. The mammography data were suggestive of malignancy BIRADS 4, which does not suggest heterogeneity or a specific type of cancer, and requires histological verification according to contemporary guidelines [46].

Concerning the differential diagnosis, LELC of the breast is a serious challenge. With the presence of poorly differentiated epithelial cells and abundant lymphocytic infiltration, it is close to the so-called “medullary” triple negative carcinoma or carcinoma of the breast with medullary characteristics. The main difference is the lack of syncytial growth patterns and the high atypia of tumor cells, characteristic of medullary carcinoma [19].

Diagnosis of LELC may be further complicated by the presence of intratumoral heterogeneity with existing foci of other types of breast carcinoma within the same tumor; the most commonly described histological variants are of lobular carcinoma [3,20,21,22].

Intratumor foci that resemble poorly differentiated ductal carcinoma are described in only one published case [23]. To the best of our knowledge, this is the first case that describes intertumor heterogeneity with LELC. It presents with a synchronous invasive ductal carcinoma in the same breast.

Lymphoproliferative diseases are also included in the differential diagnosis but are quickly ruled out based on immunohistochemical analysis. Some inflammatory conditions may also be discussed. Morphologically, LELC of the breast, as a result of dominant lymphocytic infiltration, can mimic foci with the image of lymphoepithelial lobulitis [24], abscess [25] and granulomatous mastitis [26].

While the involvement of EBV is discussed as a factor in the pathogenesis of some lymphoproliferative diseases and the low-grade nasopharyngeal carcinoma in numerous studies, it cannot be confirmed in LELC of the breast. According to Iezzoni et al. [27], EBV is only associated with LELC in the stomach, salivary glands, lungs and thymus. Pestereli et al. [21] outlined the same in 2002 and reported that the virus may be associated with tumors in the gastrointestinal and respiratory tract. Until the publication of the present case, EBV was not detected in breast LELC. The data for HPV infection are different, as positive samples were reported in three of the cases described so far [28,29,30]. Neither EBV nor HPV were tested in the presented case, since the likelihood of positivity and its clinical significance are rather negligible according to data in the literature.

Despite the data regarding a favorable course and prognosis of the lymphoepithelioma-like carcinoma of the breast, compared with aggressive nasopharyngeal carcinoma, there are nine cases that involve regional lymph nodes [21,24,25,30,31,32,33,34,35]. Distant lung metastases have been described in two cases [3,36], and there is one case of contralateral recurrence of breast LELC, which was described by Dadmanesh et al. [24]. The metastatic potential thus described confirms the need, as recapped by Ilvan et al. [37], for precise staging and assessment of the axillary status. After initial surgery in case of locoregional nodal spread, adjuvant systemic therapy may be considered. In the literature described to date, all patients underwent surgical treatment in the form of excision, quadrantectomy or mastectomy, except for one case in the work of F. Dadmanesh [24], for which there is no information. In ten of the described cases, there are data for subsequent chemotherapy [20,21,23,25,29,30,35,37,38,39], mostly combined with radiation therapy, including one case described by Nils et al. of neoadjuvant therapy, followed by radiation and adjuvant endocrine therapy [40]. The effectiveness of adjuvant radiation and chemotherapy is debatable. It has been reported as effective in case of progression or systemic recurrence [38] and distant metastasis [36].

In our case, with the presence of a synchronous non palpable invasive ductal carcinoma, we believe that organ-sparing surgery should be thoroughly considered. Initial locoregional staging with magnetic resonance should also be considered in order to determine the extent of surgery, both in the breast and the axilla. Adjuvant chemotherapy and radiation therapy may also be discussed and the effect of neoadjuvant therapy has not yet been assessed. Additionally, testing for germline mutations in BRCA 1/2 genes may also be discussed in the presence of other risk factors.

## 4. Conclusions

Breast LELC is a rare and challenging-to-diagnose tumor, in which therapeutic management is uncertain, largely based on the literature published to date. Based on the case described by us, the overall assessment of patients with LELC includes the possibility of the presence of a synchronous tumor, which is not unequivocally registered by imaging tests and could complicate subsequent therapeutic decisions and outcomes.

## Figures and Tables

**Figure 1 medicina-58-01146-f001:**
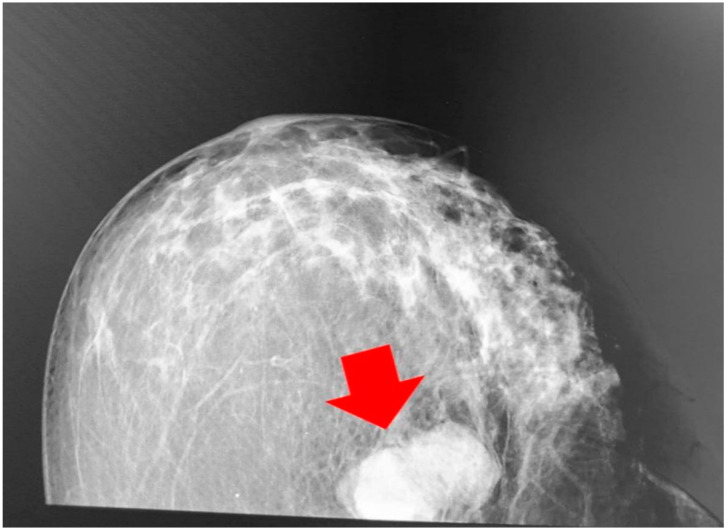
Findings from the mammography (craniocaudal position); a strong lobulated shadow can be observed (red arrow marks the breast LELC).

**Figure 2 medicina-58-01146-f002:**
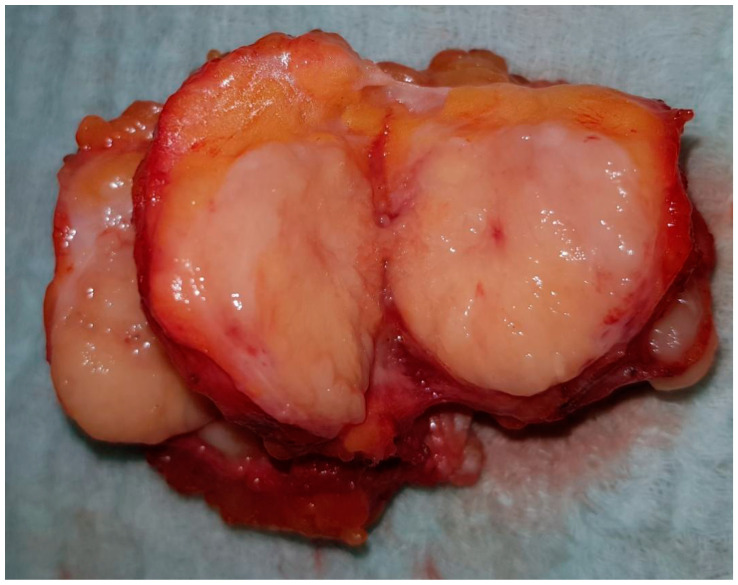
Gross specimen—non-encapsulated, circumscribed, nodular tumor.

**Figure 3 medicina-58-01146-f003:**
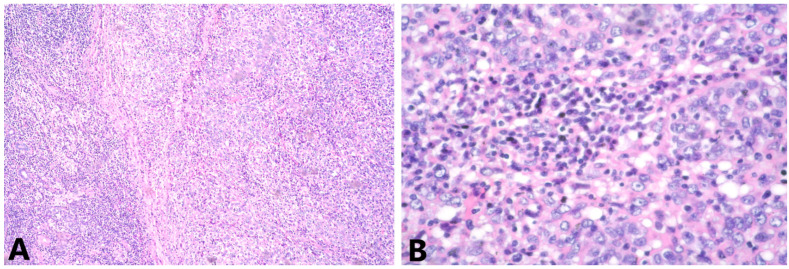
(**A**) Groups and strands of tumor cells, mixed with lymphoid stroma, HE 4x; (**B**) cells with pleomorphic vesicular nuclei, containing coarse chromatin, visible nucleoli and not abundant pale cytoplasm HE, 40x.

**Figure 4 medicina-58-01146-f004:**
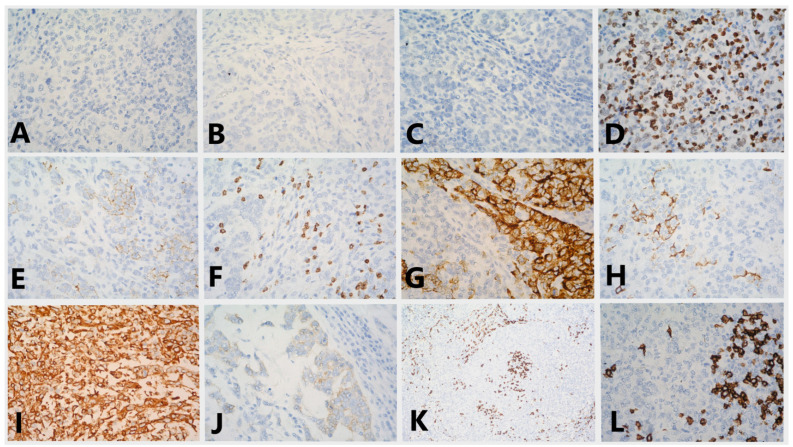
Immunostaining for (**A**) ER (lack of staining in the tumor cells, 40x); (**B**) PgR (lack of staining in the tumor cells, 40x); (**C**) HER2 (lack of staining in the tumor cells, 40x); (**D**) Ki-67 (variable staining in 50% of the tumor cells, 40x); (**E**) E-cadherin (moderately intense staining of cell membranes in 70% of the tumor cells, 40x); (**F**) p63 (nuclear positivity in about 20% of the tumor cells, 40x); (**G**) CK7 (strongly positive in about 40% of the tumor cells, 40x); (**H**) CK5/6 (weak staining in 2% of the tumor cells, 40x); (**I**) vimentin (strongly positive in the tumor cells, 40x); (**J**) CD117 (weak staining of the cytoplasm in 5% of the tumor cells, 40x); (**K**) CD3 (positive in T lymphocytes, 10x); (**L**) CD20 (positive in B lymphocytes, 40x).

**Figure 5 medicina-58-01146-f005:**
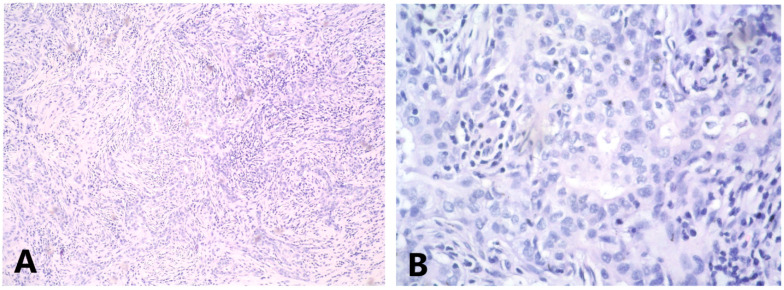
(**A**) Invasive NST ductal carcinoma G3 with prominent stromal lymphocytic infiltration HE 4x; (**B**) high power magnification of the tumor with predominantly nested growth; nuclei are high to moderately pleomorphic HE 40x.

**Figure 6 medicina-58-01146-f006:**
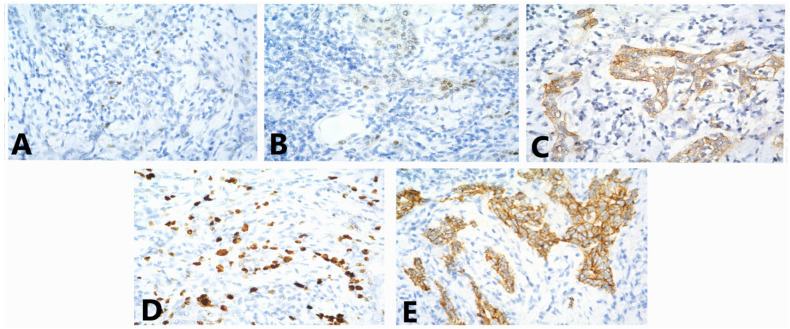
Immunostaining for (**A**) ER (weak nuclear staining in about 10% of the tumor cells, 40x); (**B**) PgR (weak nuclear staining in about 10% of the tumor cells, 40x); (**C**) HER2 (evaluated as 2+—moderately intensive, complete membrane staining, 40x); (**D**) Ki-67 index—30%, 40x; (**E**) E-cadherin (moderately intense staining of the cell membranes of the tumor cells, 40x).

## Data Availability

The authors declare that all related data are available by the corresponding author’s email.

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
