# Peer review of "Lymphoepithelioma-like Carcinoma of the Breast Synchronous with a High-Grade Invasive Ductal Carcinoma and Ductal Carcinoma in Situ in a Different Quadrant of the Same Breast: A Case Report"

_medicina, 2022, doi:10.3390/medicina58091146_

Round 1

Reviewer 1 Report

The authors present an interesting case of the very rare lymphoepithelioma-like breast cancer. With synchronous DCIS. The manuscript is well structured. However, I suggest to add more patient information and her characteristics- perhaps a table including co-morbidities and medication would help.

Further, the author should introduce the abbriviations at first use. What is NST? HER2 vs. HER 2…should be consistent.

Some spelling errors occurred throughout the manuscript (e.g. Fig. 4 and Fig 6. Immunosteining -> -staining).

Scale bars should be added to the IHC pictures. The staining procedure and results should be described in more detail. The authors can add more bpody to the mansucript by explaining why they chose the markers for staining and what results they deliver precisely. Proliferation rate and hormone receptors should be discused in terms of treatment options.

The first passage of the discussion repeats the introduction. It should refer to the presented case rather. To me it did not become clear whether the patient had EBV or not…this should be disussed as well.

Some sentences are very long and difficult to read, e.g. Page 6…Despite the data… The wording should be revized, eg. …To our knowledge… (discussion).

Reviewer 2 Report

Dear Authors:

The manuscript by Nanev et al has demonstrated the first case described in literature combining a LELC with a synchronous carcinoma in the same breast. I have just a few suggestions.

1. The linguistics need improvement

2. The background information or references about breast cancer especially LELC is missing. There are several reviews demonstrate the information about breast cancer (please cite: 1. Robot-Assisted Minimally Invasive Breast Surgery: Recent Evidence with Comparative Clinical Outcomes. doi: 10.3390/jcm11071827.    2. Advances in the Prevention and Treatment of Obesity-Driven Effects in Breast Cancers.  doi: 10.3389/fonc.2022.820968.

Best,

Round 2

Reviewer 2 Report

Strongly suggest for publication.

Author Response

Thank you for your tme